**Funding:** The author(s) received no specific funding for this work.

# Association between integrase strand transfer inhibitor (INSTIs) use with insulin resistance and incident diabetes mellitus in persons living with HIV: A systematic review and meta-analysis protocol

**Frank Mulindwa**[1]⊙*, **Habiba Kamal**[2,3]⊙, **Barbara Castelnuovo**[1‡], **Robert C. Bollinger**[4‡], **Jean-Marc Schwarz**[5‡], **Nele Brussealers**[2]⊙

**1** Infectious Diseases Institute, Makerere University, Kampala, Uganda, **2** Department of Infectious Diseases, Karolinska University Hospital, Stockholm, Sweden, **3** Department of Medicine Huddinge, Karolinska Institute, Stockholm, Sweden, **4** School of Medicine, Johns Hopkins University (JHU), Baltimore, MD, United States of America, **5** School of Medicine, University of California San Francisco, San Francisco, CA, United States of America

⊙ These authors contributed equally to this work.
‡ BC, BB and JMS also contributed equally to this work.
* mulindwafrank93@gmail.com

## Abstract

### Introduction

Poeple living with HIV have higher prevalence of diabetes mellitus and metabolic perturbations compared to non-HIV populations. Diabetes and metabolic syndrome co-morbidities add significant burden to HIV care. Currently, WHO recommends integrase strand transfer inhibitors (INSTIs) as the first or second line therapy in people with HIV due to overall good tolerability and safety profile. However, whether INSTI use increases the risk of incident diabetes (with or without metabolic syndrome) compared to other anti-retroviral therapies (ART) is controversial. In this systematic review and meta-analysis, we aim to examine this risk in HIV-positive populations receiving INSTIs compared to other ART regimens (not containing INSTIs).

### Methods and analysis

The study will be reported according to the Preferred Reporting Items for Systematic Reviews and Meta-Analysis (PRISMA) statement and the Meta-analysis Of Observational Studies in Epidemiology (MOOSE) guidelines. This protocol adheres to the Standard Protocol Items for reporting systematic reviews and meta-analyses checklist. Eligibility criteria will be original peer-reviewed published articles and conference abstracts with no language or geographical restriction; that report the ocurrence of diabetes mellitus as a discrete outcome or part of metabolic syndrome, in adult PLWHIV receiving INSTIs compared to other ART regimens. PubMed/ Medline, Web of Science, Embase and Cochrane Database of Systematic Reviews will be searched from 1st- January-2000 to 31st—January-2022. Per our a

**Competing interests:** The authors have declared that no competing interests exist.

*priori*, screening, inclusion and data extraction will be conducted separately by two investigators, and a senior researcher will be consulted in case of disagreement. The quality of included studies will be assessed by the Newcastle-Ottawa Scale (NOS) for cohort and case-control studies and the revised Cochrane risk-of-bias tool (ROB2) for randomized controlled trials. The quantitative synthesis of the study outcomes will be explored in different subgroups and sensitivity analyses. Meta regression will also be performed to further test the predictors of the outcome.

## Ethics and dissemination

Ethical approval is waived as the study is a review of published litterature. The analyses will be presented in conferences and published as a scientific article.

## Trial registrartion

PROSPERO registration number is; CRD42021273040.

## Background / Rationale

In 2016 and 2018, the World Health Organisation (WHO) recommended integrase strand transfer inhibitors (INSTIs) as preferred first and second line anti-retroviral therapy(ART) regimens respectively [1]. Subsequently, multiple countries moved to adopt dolutegravir (DTG) containing regimens with 41 (30%) of low-middle income countries (LMIC) having adopted DTG backbone regimens as the preferred first line therapy, and an additional 82 (60%) of LMIC making shifts to DTG containing regimens by Mid-2019 [2].

Despite a good safety profile and high tolerability, INSTIs have been linked in some reports to various metabolic abnormalities including new onset diabetes mellitus (DM) [3–6]. The hypothesized pathophysiology has been the medication's ability to chelate magnesium at cellular level which in turn affects glucose transport via glucose transporter type 4 (GLUT-4) receptors causing insulin resistance (IR) [7]. Furthermore, INSTIs have been shown to cause weight gain in patients, which in turn also can contribute to insulin resistance and eventual development of diabetes mellitus [8–12]. These theories may however not conclusively explain multiple case reports of patients presenting with accelerated hyperglycaemia after initiation of DTG, bictegravir and raltegravir with preceding weight loss [3, 5, 13].

Data from observational studies associating INSTIs use and diabetes mellitus have been contradictory. In a large North American cohort, a 22% higher risk of diabetes was demonstrated in patients on INSTIs and protease inhibitors (PI) than those on non- nucleoside reverse transcriptase inhibitors (NRTS) [6], while results from a large French cohort did not show any association [14].

To the best of our knowledge, one narrative systematic review has been published evaluating the association between integrase inhibitors and diabetes mellitus as a subset of metabolic syndrome [8]. Pooled studies in that narrative review were suggestive of a higher risk of diabetes and metabolic syndrome in patients on INSTI versus other HIV drug classes. In our systematic review and meta-analysis, we aim to pool data from randomised controlled trials, case-control and cohort studies with clear temporal causality so as to ascertain and quantify this association.

## Aim

To assess the association between the use of integrase strand transfer inhibitors in people living with HIV with new onset type 2 diabetes mellitus and insulin resistance compared to other ART regimens.

## Objectives and hypothesis

### Objective

To assess the risk of developing insulin resistance (IR) and new onset diabetes mellitus, as a discrete outcome or part of metabolic syndrome in HIV positive populations treated with INSTIs compared to other ART regimens.

### Hypothesis

We hypothesize that use of INSTIs is associated with increased insulin resistance and diabetes mellitus compared to other ART regimens.

## Methods

The Preferred Reporting Items for Systematic Reviews and Meta-analyses (PRISMA-7) [15] and the Meta-analysis Of Observational Studies in Epidemiology (MOOSE) [16] guidelines will be followed in the execution and reporting of this systematic review and meta-analysis. This protocol adheres to the Standard Protocol Items for reporting systematic reviews and meta-analyses checklist [17].

## Inclusion criteria

**Time interval.** Published peer-reviewed articles and conference abstracts between 1st-January 2000 to 31st- January-2022 will be considered. We shall consider studies from 2000 so as to capture phase III trials given the first INSTI was FDA approved in 2007 [18].

**Study designs.** Randomized controlled trials, prospective and retrospective cohort studies, and case-control studies in English without geographical restriction.

**Study setting.** single, multi-centre, population-based and different levels of care will be included. Studies will be selected without geographical restrictions.

**Study participants.** HIV positive individuals with no diabetes and/or metabolic syndrome at study entry, receiving INSTIs compared to patients on other ART regimens. We will include both ART experienced patients switching to integrase inhibitors and ART naïve patients newly starting integrase inhibitors.

**Exposure and comparator group.** Exposure will be defined as taking any of the INSTIs for $\geq$ 12 weeks i.e. raltegravir, elvitegravir, dolutegravir, bictegravir, and cabotegravir, without protease inhibitors or non-nucleoside reverse transcriptase inhibitors, independent of additional ART products. Comparator groups will be patients on protease inhibitors and/or non-nucleoside transcriptase inhibitors.

## Study outcome

Studies with the following outcome definitions will be included (Table 1):

Diabetes: WHO [19] or American Diabetes Association (ADA) diagnostic criteria [20] or need for diabetes medication.

Metabolic syndrome: according to International Diabetes Federation (IDF) [21] or World Health Organisation (WHO) [22] diagnostic criteria.

**Table 1. Definition of study outcome measures.**

| Study outcome | Acceptable outcome measures in individual studies for inclusion in the meta-analysis |
|---|---|
| Diabetes mellitus | ADA criteria [24]: HbA1C ≥6.5% or FPG ≥126 mg/dL (7.0 mmol/L) or 2-h PG ≥200 mg/dL (11.1 mmol/L) during an OGTT |
| | WHO criteria [19]: |
| | Fasting plasma glucose values of ≥ 7.0 mmol/L (126 mg/dl) OR |
| | 2-h post-load plasma glucose ≥ 11.1 mmol/L (200 mg/dl) OR |
| | HbA1c ≥ 6.5% (48 mmol/mol) OR |
| | Random blood glucose ≥ 11.1 mmol/L (200 mg/ dl) in the presence of signs and symptoms. |
| | Need for diabetes medication |
| Metabolic syndrome | 1-NCEP ATP III criteria [25]: The presence of three or more of the following risk determinants: |
| | Increased waist circumference (>102 cm [>40 in] for men, >88 cm [>35 in] for women); |
| | Elevated triglycerides (≥150 mg/dl); |
| | Low HDL cholesterol (<40 mg/dl in men, <50 mg/dl in women); |
| | Hypertension (≥130/≥85 mmHg); and |
| | Impaired fasting glucose (≥110 mg/dl) |
| | 2-WHO criteria [22]: Glucose intolerance, DM2 or insulin-resistance in addition to at least two of the following: |
| | BMI > 30 and HWR > 0.9 (M) and > 0.85 (F) |
| | Serum TG ≥ 150mg/dl |
| | Serum HDL < 35mg/dl (M), <39mg/dl (F) |
| | Blood pressure ≥ 140/90 or on hypertension treatment |
| | Other risk factors: microalbuminuria ≥20mcg/min |
| | 3-IDF [21]: DM/ Glucose intolerance and two or more criteria |
| | Fasting glucose of 100-125mg/dl Or DM 2 |
| | WC ≥ 94cm (M), 80cm (F) |
| | TG ≥150mg/dl |
| | HDL <40mg/dl or <50mg/dl |
| | On treatment for SAH/ BP ≥130/85mmHG |
| | 4-European Group for Study of Insulin Resistance definition [26]: Elevated plasma insulin (>75th percentile) plus two other factors from among the following: |
| | Abdominal obesity: waist circumference (WC) ≥94 cm in men and ≥80 cm in women |
| | Hypertension: ≥140/90 mm of Hg or on antihypertensive treatment |
| | Elevated triglycerides (≥150 mg/dl) and/or reduced HDL-C (<39 mg/dl for both men and women) |
| | Elevated plasma glucose: impaired fasting glucose (IFG) or IGT, but no diabetes |
| Insulin resistance | Homeostatic model for Insulin resistance (HOMA-IR) [23] |

ADA- American Diabetes Association, HbA1C- Glycated Hemoglobin, PG- Plasma Glucose, OGTT- Oral Glucose Tolerance test, WHO- World Health Organization, NCEP ATP III- National Cholesterol Education Program Adult Treatment Panel III, HDL- High Density Lipoproteins, DM2- Diabetes Mellitus type II, BMI- Body Mass Index, TG- triglycerides, IDF-International Diabetes Federation, WC- Waist Circumference, SAH- Systemic Arterial Hypertension, BP- Blood Pressure, IFG- Impaired Fasting Glucose, IGT- Impaired Glucose Tolerance, HOMA-IR- Homeostatic Model for Insulin Resistance.

Insulin resistance: Homeostatic model for insulin resistance (HOMA-IR) [23].

**Outcome measures.** Studies reporting; diabetes mellitus relative risks, mean changes in blood glucose or mean changes in HOMA-IR. Adjusted relative risks will be sought whenever available, otherwise raw data to calculate unadjusted effect estimates will be included.

## Exclusion criteria

The following studies will be excluded:

1. Cross-sectional, prevalence analysis and self-report of outcomes.

2. Non-HIV populations, pregnant individuals, assessing the outcome in the same population at different time intervals.

3. Studies reporting sole outcomes as changes of weight or fat distribution.

4. Studies reporting the effect of interventional exercise or drugs on metabolic parameters (anti-diabetes drugs, statins or other lipid lowering drugs)

5. Studies analysing the same cohort (studies with eligible design and with the most complete set of results will be considered).

6. Studies lacking data to compute risk estimates (authors will be contacted).

## Source of information and search strategy

A systematic literature search will be conducted by Karolinska Institute librarians (S1 Checklist). The following databases will be searched: Pubmed/Medline, Embase, and Web of science databases and results will be managed in endnote library. After removal of duplicate citations, screening by title and abstract will be conducted to sort articles for full review. If data is overlapping, the study with the longest follow-up period and the most extractable data will be included. For the screened studies, backward and forward citation tracking will be performed to identify additional studies. Reviews and editorials will be searched for relevant literature. Grey literature will be searched using the Cochrane Central Register of Controlled Trials (CENTRE) as well as conference proceedings of the major international HIV conferences, for published studies that meet the study *a priori*.

## Data extraction

**Data items.**  The following data items will be collected independently by FM and HK in excel sheets with disagreements resolved by a senior reviewer, NB

Study title, author, year of publication, journal, country, design, setting, study period, inclusion and exclusion criteria, follow-up duration, baseline demographic and clinical characteristics, outcome measures at baseline and at end of follow up. For unavailable data, we will contact authors for supplemental data

**Risk of bias and study quality assessment.**  Publication bias will be assessed by Egger's test via funnel plot symmetry assessment for all the included studies. Quality assessment of individual studies will be carried out by two reviewers and a senior reviewer will help solve disagreements. For included RCTs, the Cochrane risk of bias tool for RCTs [27] and the Newcastle Ottawa scale for cohort studies [28] will be used. Results will be reported in a table form for easy interpretation. We will not exclude studies based on the quality scores but rather perform sensitivity analysis to evaluate the effect of individual study quality on the final results of the meta-analysis.

## Data synthesis

Incidence rates or hazard ratios for incident diabetes with corresponding 95%CIs will be pooled together in a random effects model. In studies reporting mean changes in blood glucose and HOMA-IR, means with corresponding 95% confidence intervals (95%CIs) will be calculated in a random effects model. The most adjusted means will be considered.

Forest plots will be created to visually inspect and analyse the effect sizes and corresponding 95% CI. If possible, subgroup analysis or meta-regression (if more or equal to 10 studies) will be performed for crude and adjusted estimates, particular integrase inhibitor, study designs, sex, age, duration of follow-up, geographical origin of the study population, ART status on enrolment into the study (first- or second-line treatment), and other HIV-related factors to assess contribution to effect sizes.

Heterogeneity across studies will be assessed with Higgins's $I^2$ test [29]. $I^2 > 75\%$ will be considered indicative of high statistical heterogeneity, $>50\%$ of moderate and $<25\%$ of low statistical heterogeneity across studies.

## Confidence in cumulative evidence

This study will assess the available evidence regarding the association of the use of INSTIs and development of diabetes mellitus. It will additionally explore a possible causative hypothesis i.e. insulin resistance, towards the development of diabetes. Previous systematic reviews about the association of INSTIs and diabetes have largely been narrative. The proposed meta-analysis will bridge this gap.

Per our a priori, we are selecting robust study designs with clear causal inferences i.e. RCTs, nested case control studies with baseline IR/ blood glucose measurements and cohort studies which will enhance pooled study quality.

## Supporting information

**S1 Checklist. Performance of the proposed systematic review protocol on the PRISMA-P (Preferred Reporting Items for Systematic review and Meta-Analysis Protocols) 2015 checklist.**
(DOC)

## Acknowledgments

We acknowledge Karolinska Institute librarians; GunBrit Knutssön & Narcisa Hannerz for performing the preliminary data search.

## Author Contributions

**Conceptualization:** Frank Mulindwa, Jean-Marc Schwarz, Nele Brussealers.

**Investigation:** Frank Mulindwa.

**Methodology:** Frank Mulindwa, Habiba Kamal, Nele Brussealers.

**Supervision:** Robert C. Bollinger, Jean-Marc Schwarz, Nele Brussealers.

**Visualization:** Habiba Kamal.

**Writing – original draft:** Frank Mulindwa, Habiba Kamal.

**Writing – review & editing:** Frank Mulindwa, Habiba Kamal, Barbara Castelnuovo, Robert C. Bollinger, Jean-Marc Schwarz, Nele Brussealers.

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
