## [Decision Letter · Decision Letter 0]

10 Jan 2022

PONE-D-21-34754Association between integrase strand transfer inhibitor (INSTIs) use with insulin resistance and incident diabetes mellitus in persons living with HIV; A systematic review and meta-analysis protocolPLOS ONE

Dear Dr. Frank Mulindwa,

Thank you for submitting your manuscript to PLOS ONE. After careful consideration, we feel that it has merit but does not fully meet PLOS ONE’s publication criteria as it currently stands. Therefore, we invite you to submit a revised version of the manuscript that addresses the points raised during the review process.

We look forward to receiving your revised manuscript.

Kind regards,

Surasak Saokaew, PharmD, PhD, BPHCP, FACP

Academic Editor

PLOS ONE

Journal Requirements:

3. We note you have included a table to which you do not refer in the text of your manuscript. Please ensure that you refer to Table 1 in your text; if accepted, production will need this reference to link the reader to the Table.

Reviewers' comments:

Reviewer's Responses to Questions

**Comments to the Author**

1. Does the manuscript provide a valid rationale for the proposed study, with clearly identified and justified research questions?

Reviewer #1: Yes

Reviewer #2: Yes

2. Is the protocol technically sound and planned in a manner that will lead to a meaningful outcome and allow testing the stated hypotheses?

Reviewer #1: Partly

Reviewer #2: Yes

3. Is the methodology feasible and described in sufficient detail to allow the work to be replicable?

Reviewer #1: Yes

Reviewer #2: Yes

4. Have the authors described where all data underlying the findings will be made available when the study is complete?

Reviewer #1: Yes

Reviewer #2: Yes

5. Is the manuscript presented in an intelligible fashion and written in standard English?

Reviewer #1: Yes

Reviewer #2: Yes

6. Review Comments to the Author

You may also provide optional suggestions and comments to authors that they might find helpful in planning their study.

Reviewer #1: This is an important study and the manuscript reads well. Overall it is technically sound. I did have a number of queries and considerations which may be pertinent.

1. Could the authors explain how they will account for prior ART exposure, and previous ART regimens

2. The period of study is over a large time period January 2000 to May 2021. There have been several clinical management developments during this time which may have a bearing. Have the authors considered this and might a shorter time period of review be more advantageous in this respect.

3. It is difficult to define a priori sub group analysis but some ideas of potential areas may enhance the protocol. Amongst this geographical and ethnic subgroups might provide useful insights.

Reviewer #2: It's an interesting study. There are no studies evaluating the association between Integrase inhibitor and diabetes or insulin resistance.

The authors should use official abbreviation for drug group name.

7. PLOS authors have the option to publish the peer review history of their article (what does this mean?). If published, this will include your full peer review and any attached files.

Reviewer #1: No

Reviewer #2: No

---

## [Author Response · Author response to Decision Letter 0]

29 Jan 2022

Reviewers’ comments:

Reviewer #1: This is an important study and the manuscript reads well. Overall, it is technically sound. I did have a number of queries and considerations which may be pertinent.

1. Could the authors explain how they will account for prior ART exposure, and previous ART regimens

>> Reply: We plan to perform a sub -group analysis to evaluate the effect on integrase inhibitors on incident diabetes and insulin resistance in ART naïve patients and ART exposed patients. This is highlighted on page 9 (data synthesis, lines; 196-201; Forest plots will be created to visually inspect and analyse the effect sizes and corresponding 95% CI. If possible, subgroup analysis or meta-regression (if more or equal to 10 studies) will be performed for crude and adjusted estimates, particular integrase inhibitor, study designs, sex, age, duration of follow-up, geographical origin of the study population, ART status on enrolment into the study (first- or second-line treatment), and other HIV-related factors to assess contribution to effect sizes.

However, we agree with the reviewer that certain types of ART (previous exposure) may confound the present risk to diabetes more than other types. We have included this in the summary page- page 2- limitations

2. The period of study is over a large time period January 2000 to May 2021. There have been several clinical management developments during this time which may have a bearing. Have the authors considered this and might a shorter time period of review be more advantageous in this respect.

>> Reply: We used that timeline so as to capture phase III trials comparing efficacy of integrase inhibitors as compared to protease inhibitors and non-nucleoside reverse transcriptase inhibitors. The first integrase inhibitor, Raltegravir was FDA approved in 2007 and how far backwards we searched was informed by a preliminary data search. We agree with the reviewer that over that time, various advances have been made with integrase inhibitors including the introduction of dolutegravir, elvitegravir, bictegravir and carbotegravir. We hope to account for this by performing sub group analysis according to the individual integrase inhibitor as highlighted on page 9 (data synthesis, lines; 196-201). This would help delineate whether newer generation drugs in that class are different from the older drugs in terms of glucose metabolism.

Additionally to capture new studies which would have been published, we have extended our search timeline from May 2021 to 31st – January – 2022 as highlighted on page 3, line 57-62 and page 5, lines 118-120.

3. It is difficult to define a priori sub group analysis but some ideas of potential areas may enhance the protocol. Amongst this geographical and ethnic subgroup might provide useful insights.

>> Reply: we added these to the protocol as follows (page 9, line 197-201); If possible, subgroup analysis or meta-regression (if more or equal to 10 studies) will be performed for crude and adjusted estimates, particular integrase inhibitor, study designs, sex, age, duration of follow-up, geographical origin of the study population, ART status on enrolment into the study (first- or second-line treatment), and other HIV-related factors to assess contribution to effect sizes.

Reviewer #2: It's an interesting study. There are no studies evaluating the association between Integrase inhibitor and diabetes or insulin resistance.

The authors should use official abbreviation for drug group name.

>> Reply: Thank you very much for the comment. We found the two acceptable abbreviations for HIV integrase inhibitors are: INI- integrase inhibitors and INSTI- integrase strand transfer inhibitors. In our manuscript we used ‘INSTI’.

---

## [Editor Report · Decision Letter 1]

17 Feb 2022

Association between integrase strand transfer inhibitor (INSTIs) use with insulin resistance and incident diabetes mellitus in persons living with HIV: A systematic review and meta-analysis protocol.

PONE-D-21-34754R1

Dear Dr. Frank Mulindwa

We’re pleased to inform you that your manuscript has been judged scientifically suitable for publication and will be formally accepted for publication once it meets all outstanding technical requirements.

Kind regards,

Surasak Saokaew, PharmD, RPh, PhD, BPHCP, FACP, FCPA

Academic Editor

PLOS ONE
---

## [Editor Report · Acceptance letter]

21 Feb 2022

PONE-D-21-34754R1 

Association between integrase strand transfer inhibitor (INSTIs) use with insulin resistance and incident diabetes mellitus in persons living with HIV: A systematic review and meta-analysis protocol.

Dear Dr. Mulindwa:

I'm pleased to inform you that your manuscript has been deemed suitable for publication in PLOS ONE. Congratulations! Your manuscript is now with our production department. 

Kind regards, 

on behalf of

Dr. Surasak Saokaew 

Academic Editor

PLOS ONE